# Redundant Queries in DETR-Based 3D Detection Methods: Unnecessary and Prunable

## Abstract

Query-based models are extensively used in 3D object detection tasks, with a wide range of pre-trained checkpoints readily available online. However, despite their popularity, these models often require an excessive number of object queries, far surpassing the actual number of objects to detect. The redundant queries result in unnecessary computational and memory costs. In this paper, we find that not all queries contribute equally – a significant portion of queries have a much smaller impact compared to others. Based on this observation, we propose an embarrassingly simple approach called **G**radually **P**runing **Q**ueries (GPQ), which prunes queries incrementally based on their classification scores. A key advantage of GPQ is that it requires no additional learnable parameters. It is straightforward to implement in any query-based method, as it can be seamlessly integrated as a fine-tuning step using an existing checkpoint after training. With GPQ, users can easily generate multiple models with fewer queries, starting from a checkpoint with an excessive number of queries. Experiments on various advanced 3D detectors show that GPQ effectively reduces redundant queries while maintaining performance. Using our method, model inference on desktop GPUs can be accelerated by up to 1.31x. Moreover, after deployment on edge devices, it achieves up to a 67.86% reduction in FLOPs and a 76.38% decrease in inference time. The code will be available soon.

## 1 Introduction

3D object detection is a key task for autonomous driving. Among various algorithms, DETR-based methods (Carion et al., 2020; Wang et al., 2022; Liu et al., 2022; Wang et al., 2023a;b) stand out for their end-to-end detection capabilities without relying on hand-crafted components, thanks to their set-prediction pipeline. A key feature of DETR-based models is the use of pre-defined queries in transformer modules, which are generated from pre-defined reference points Zhu et al. (2020); Liu et al. (2022; 2023a); Wang et al. (2023a;b); Jiang et al. (2024); Li et al. (2022); Yang et al. (2023); Liao et al. (2023a;b). These queries are refined in the self-attention module and interact with image features in the cross-attention module. The updated queries are then passed through MLPs to predict classification scores and 3D bounding boxes.

Despite their effectiveness, these methods are computationally intensive due to the large number of object queries required. More precisely, the number of pre-defined queries is typically set to 300 for 2D object detection tasks (Meng et al., 2021; Chen et al., 2023; Gao et al., 2022), and this number increases to 900 for 3D detection (Wang et al., 2022; Li et al., 2023; Liu et al., 2022; Wang et al., 2023a;b; Jiang et al., 2024), which significantly exceeds the actual number of objects in both cases. As depicted in DETR3D (Wang et al., 2022), the performance of the model is positively correlated with the number of queries. We conducted experiments with different query configurations of StreamPETR (Wang et al., 2023b), which further validates this conclusion. Specifically, as shown in Table 1, the model's performance consistently declines as the number of queries is reduced.

Since the number of predictions during model inference is tied to the number of queries, fewer queries lead to fewer predictions. Reducing queries requires additional computation to cover the solution space and may even result in failure to find optimal parameters. Therefore, using fewer queries generally results in poorer performance. For instance, as illustrated in Figure 1 (a), if a model were to use only a single query to predict two object types – such as pedestrians and barriers

| # Ref. Queries | # Pro. Queries | # Tot. Queries | mAP↑ | NDS↑ | Memory(MiB)↓ | FPS↑ |
|---|---|---|---|---|---|---|
| 1288 | 512 | 1800 | 39.62% | 0.4965 | 2580 | 12.7 |
| 1074 | 426 | 1500 | 39.37% | 0.4945 | 2520 | 15.7 |
| 858 | 342 | 1200 | 39.23% | 0.4960 | 2466 | 15.8 |
| 644 | 256 | 900 | 37.83% | 0.4737 | 2338 | 16.1 |
| 430 | 170 | 600 | 37.43% | 0.4770 | 2334 | 17.0 |
| 236 | 64 | 300 | 33.62% | 0.4429 | 2332 | 18.5 |
| 108 | 42 | 150 | 26.46% | 0.3763 | 2332 | 18.8 |
| 64 | 26 | 90 | 19.54% | 0.2940 | 2330 | 18.8 |

Table 1: Results of StreamPETR (Wang et al., 2023b) with varying numbers of queries. The model utilizes two types of queries when inference: queries generated from pre-defined reference points (Ref. Queries) and queries propagated from Ref. Queries (Pro. Queries). All experiments were conducted over a total of 24 epochs.

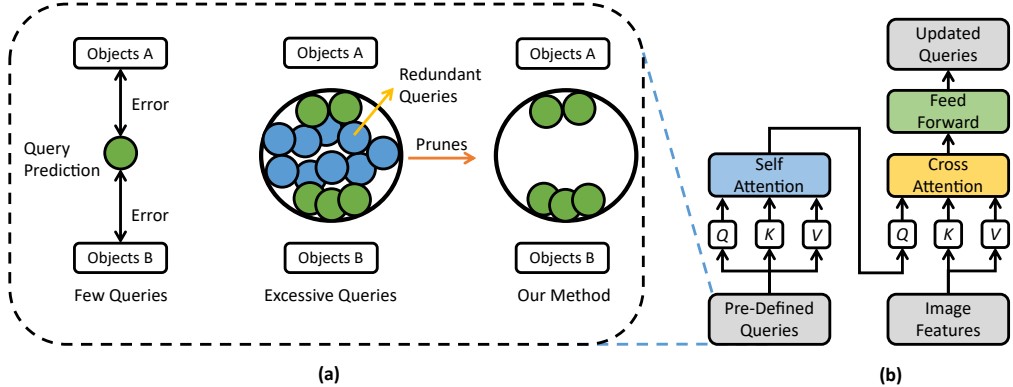

Figure 1: Illustration of query-based detection methods. (a) When using only a few queries, they must balance across different object instances, leading to poor prediction performance. Introducing excessive queries allows each one to handle a specific object instance, but this creates redundancy. Our method removes redundant queries that contribute little to the model's performance. (b) The workflow of a single transformer layer. Pre-defined queries are fed into the self-attention module, where they interact with each other. The output of self-attention then serves as the *query* in cross-attention, with image features acting as the *key* and *value*.

– that query would need to capture the features of both objects simultaneously. By handling multiple tasks, the query would need to strike a compromise between the performance of the two object types. If we use more queries, such as two, one can focus on detecting pedestrians while the other identifies barriers, which reduces the interference between two queries. This means that the more queries there are, the less burden each individual query has to bear. Given the complexity of spatial attributes in 3D detection tasks – such as location, rotation, and velocity for moving objects – it is understandable that the performance of query-based models is sensitive to the number of queries.

## 1.1 CONFIGURING AN EXCESSIVE NUMBER OF QUERIES IS INEFFICIENT

As is well known, DETR-based models use the Hungarian algorithm to match predictions with ground truth labels. When the number of queries significantly exceeds the number of objects to be detected, most predictions are treated as negative instances during bipartite matching. In fact, in most non-specialized scenes, the number of objects to be detected is typically fewer than 100 (Caesar et al., 2020; Wilson et al., 2021), implying that during the training process, the ratio of negative to positive instances could reach 8:1. For these negative instances, there are no ground truth labels to serve as supervision. As a result, during loss calculation, zero vectors are matched with negative predictions, progressively driving these queries to produce lower classification scores.

In cases where positive instances far outnumber negative ones, the selection of queries would gradually become imbalanced. This not only results in wasted computational resources but also causes the classification scores of queries that are more frequently selected as negative instances to be significantly lower than others. During inference, NMS-free selector directly selects predictions with

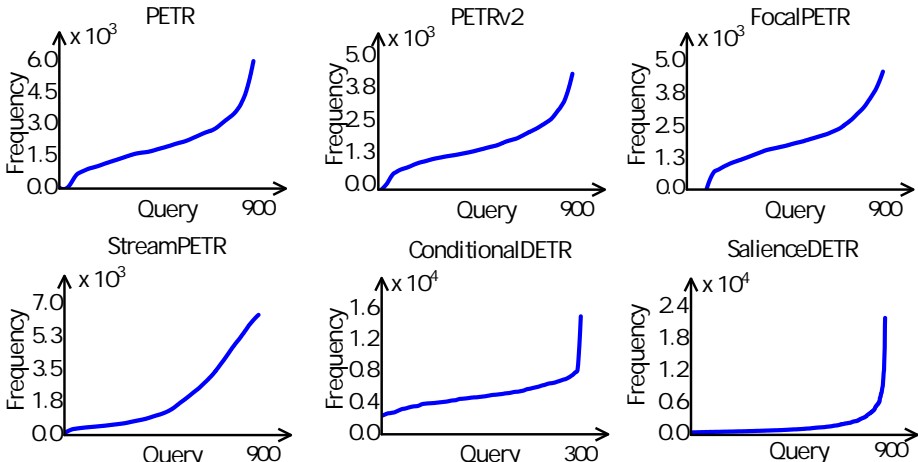

Figure 2: Selection frequency of queries (sorted in ascending order) during inference for different methods. Among these methods, PETR, PETRv2, FocalPETR and StreamPETR are 3D object detection methods, and the other two are 2D object detection methods. As illustrated, the selection frequency of queries is imbalanced across both 2D and 3D query-based methods. In PETR, PETRv2, and FocalPETR, there are even queries that were never selected as final results.

the highest classification scores as final results(Carion et al., 2020). For instance, if we use 900 queries to predict 10 classes, the model generates a tensor of shape $900 \times 10$, which corresponds to $900 \times 10 = 9000$ instances. The top-$k$ instances with the highest classification scores are selected as final predictions. As a result, queries with lower classification scores are unlikely to be selected. As illustrated in figure 2, during inference, the frequency at which queries are selected becomes imbalanced. Even some queries are never selected at all. These underutilized queries produce background predictions and contribute far less to the model's output than the more frequently selected queries.

Given that these queries play a minimal role, is there a way to discard them without compromising the model's performance?

### 1.2 PRUNING REDUNDANT QUERIES

To address query redundancy, we propose GPQ, a method that gradually prunes redundant queries. Queries with lower classification scores are considered less contributive to detection and exhibit weaker representation capabilities. As a result, we remove these lower-scoring queries, retaining only those with higher classification scores to ensure better performance.

Our method is embarrassingly simple: load a checkpoint and run the model as it should be, then sort the queries by classification scores after each iteration, and remove the bottom-$k$ queries every $n$ iterations. Compared to existing pruning methods that utilize a learnable binary mask (Yu et al., 2022; Kong et al., 2022; Tang et al., 2023; Shi et al., 2023; Chen et al., 2024; Ilhan et al., 2024; Khaki & Plataniotis, 2024; Wei et al., 2023; Yu & Xiang, 2023), GPQ introduces no additional learnable parameters and, therefore, incurs no extra computational cost. Additionally, our method allows for the creation of multiple model versions with varying numbers of queries using an existing checkpoint that contains a large number of queries. This eliminates the need to re-train the model with fewer queries, which would require additional time to restore performance.

Our contributions can be summarized as follows:

1. To the best of our knowledge, we are the first to address the issue of redundant queries in commonly used 3D object detectors and to conduct a comprehensive analysis of the role of queries in detection transformers. Our findings indicate that the majority of queries in existing query-based methods are redundant and unnecessary.

2. We propose an embarrassingly simple yet effective strategy that gradually prunes redundant queries in detection transformers, enabling us to better utilize existing pre-trained checkpoints to reduce model complexity while maintaining detector performance.

3. We conducted extensive experiments on various query-based detectors to evaluate the effectiveness of our proposed method. The results indicate that, on desktop GPUs, GPQ achieves inference acceleration of up to 1.31× and it achieves at most a 67.87% reduction in FLOPs with a 76.38% decrease in inference time after deployment on edge devices.

## 2 RELATED WORK

### 2.1 DETR-BASED 3D OBJECT DETECTORS

Based on transformers (Vaswani et al., 2017), DETR-based methods have been widely used in 3D object detection tasks (Li et al., 2022; Yang et al., 2023; Liu et al., 2022; 2023a; Wang et al., 2023a; Jiang et al., 2024; Li et al., 2023; Liu et al., 2023b; Chen et al., 2022; Wang et al., 2024b; Xie et al., 2023). Liu et al. (2022) develops position embedding transformation to generate position-aware 3D features. Liu et al. (2023a) introduces temporal modeling and generates 3D positional embedding according to features of input images. Wang et al. (2023b) proposes an object-centric temporal modeling method that combines history information with little memory cost. By utilizing high-quality 2D object prior, Far3D (Jiang et al., 2024) generates 3D adaptive queries to complement 3D global queries, which extends the detection range to 150 meters.

All these methods use similar pre-defined queries, either as learnable parameters or generated from pre-defined reference points. Despite variations in sampling strategies (Li et al., 2022; Zhu et al., 2020), they all share the *query*, *key* and *value* components, which are processed through transformer layers. While these methods share similar designs and advantages, they also face common drawbacks: high computational costs and excessive memory usage. This makes it particularly challenging to deploy DETR-based methods on edge devices.

### 2.2 PRUNING TRANSFORMERS

Distillation and pruning methods have been developed to reduce the resource requirements of large models (Yang et al., 2022; Li et al., 2024; Liang et al., 2023). As transformers have become increasingly prominent in fields like natural language processing and computer vision, numerous pruning methods targeting transformer models have been proposed (Michel et al., 2019; Fan et al., 2019; Chen et al., 2021; Sanh et al., 2020; Liu et al., 2024; Lagunas et al., 2021; Yu et al., 2022; Kwon et al., 2022; Yu & Xiang, 2023). In particular, Michel et al. (2019) discovers that a large percentage of heads can be removed at test time. Fan et al. (2019) randomly drops transformer layers at training time. Chen et al. (2021) explores sparsity in vision transformers, which proposes an unstructured and a structured methods to prune the weights. While magnitude pruning methods rely on the absolute values of parameters to determine which units to prune, Sanh et al. (2020) uses first-order information rather than zero-order information as the pruning criterion. Lagunas et al. (2021) extends Sanh et al. (2020) to local blocks of any size. Yu et al. (2022) reduces both width and depth of transformer simultaneously. Kwon et al. (2022) proposes a post-training pruning method that does not need retraining. Yu & Xiang (2023) proposes an explainable method by assigning each unit an explainability-aware mask. Liu et al. (2024) prunes tokens in ViT(Dosovitskiy, 2020) model, where pruned tokens in the feature maps are preserved for later use.

Most of the aforementioned pruning methods use a binary mask to determine which parameters to prune (Yu & Xiang, 2023; Yu et al., 2022; Sanh et al., 2020; Lagunas et al., 2021), which adds extra training costs. Furthermore, these methods often struggle to adapt effectively to 3D object detection models or show limited performance when applied to such tasks. This limitation arises from several key challenges:

**Nonexistent Pruning Targets.** Many pruning methods for NLP or ViT models focus on "attention head" pruning (Michel et al., 2019; Kwon et al., 2022; Yu et al., 2022). This is feasible because, in these models, "attention heads" are well-defined, trainable structural components that can be partially pruned using techniques like masking. However, in object detection methods, particularly 3D object detection, attention heads are essentially implemented as reshaping operations (Wang et al., 2022; Liu et al., 2022; 2023a; Wang et al., 2023a). Modifying their number does not impact the computational cost. Consequently, such pruning methods are not applicable to 3D object detection.

**Structural Inconsistencies.** In NLP or ViT models, an important assumption is that tokens generated from the input simultaneously act as queries, keys, and values. This ensures the resulting attention map is a square matrix with dimensions $N_q \times N_k$. Several pruning methods depend on this square attention map structureWang et al. (2024a). However, in object detection tasks, predefined queries are commonly used, resulting in $N_q \neq N_k$. As a result, the attention map becomes a non-square matrix, rendering these methods unsuitable for 3D object detection models.

**Massive Data Differences.** In ViT models, each batch typically processes a single image, producing fewer than 200 tokens per image. However, in 3D object detection, the need to predict additional indicators and process multi-view images significantly increases the token count. Even with lower resolutions, at least 4,000 tokens are generated, and this number can exceed 16,000 when using larger backbones and higher resolutions. The sheer volume of tokens presents a significant challenge for applying token-pruning methods to 3D object detection, as the computational cost of these methods often scales with the number of tokens(Bolya et al.; Wang et al., 2024a).

These challenges highlight the need for dedicated pruning strategies tailored to the unique characteristics of 3D object detection models.

## 3 METHOD

### 3.1 TRANSFORMER AND PRE-DEFINED QUERIES REVISITING

The scaled-dot attention (Vaswani et al., 2017) operation contains three inputs: *query* $Q \in \mathbb{R}^{N_q \times E}$, *key* $K \in \mathbb{R}^{N_k \times E}$ and *value* $V \in \mathbb{R}^{N_k \times D_v}$. An attention weight matrix will be calculated using *query* and *key*, which will be used to sample *value*. Learnable parameters in attention operation are its projection matrices $W^Q \in \mathbb{R}^{E \times E}, W^K \in \mathbb{R}^{E \times E}, W^V \in \mathbb{R}^{D_v \times D_v}$ and $W^O \in \mathbb{R}^{D_v \times D_v}$:

$$\text{AttnOut} = \text{Softmax}\left(\frac{(QW^Q)(KW^K)^{\text{T}}}{\sqrt{D_v}}\right)(VW^V) \cdot W^O \in \mathbb{R}^{N_q \times D_v} \tag{1}$$

where $N_q$ is the number of queries, $N_k$ is the number of keys and values, $E$ is the dimension for *query* and *key*, and $D_v$ is the dimension for *value*.

Each transformer layer used in query-based methods typically consists of a self-attention layer, a cross-attention layer, and a feedforward network, as illustrated in Figure 1 (b). The self-attention module uses the pre-defined queries as *query*, *key*, and *value*, and its output is then fed into the cross-attention module as the *query* to interact with image features.

To stack multiple transformer layers, the output of layer $l$ serves as the input for layer $l+1$, allowing the pre-defined queries to be updated. Let $F_I \in \mathbb{R}^{N_k \times E}$ represent the image features, this process can be described as follows:

$$Q_l \leftarrow \begin{cases} \text{Self-Attention}(Q_l) & , l = 0 \\ \text{Self-Attention}(Q_{l-1}) & , l > 0 \end{cases} \tag{2}$$

$$Q_l \leftarrow \text{Cross-Attention}(Q_l, F_I, F_I) \tag{3}$$

$$Q_l \leftarrow \text{FFN}(Q_l) \tag{4}$$

where $l$ is the index of current transformer layer, and FFN is the feedforward network, which is a multi-layer perception with a large hidden dimension $h \gg D_v$. $Q_0$ represents the initial pre-defined queries, which will be updated at each layer through interactions with themselves in self-attention and with image features in cross-attention.

### 3.2 THE ALGORITHM OF GRADUALLY PRUNING QUERIES

We consider each query as the fundamental unit for pruning and **use classification scores as the pruning criterion**. During iteration process, we gradually prune redundant queries. The pruning operation is triggered every $n$ iterations. Each time, we select the query that generates the lowest classification scores and remove it from current queries. The dropped query is immediately removed from the model. During both training and inference, these queries will no longer participate in any operations. Note that it is the queries, not the predictions, that are dropped. The pruning procedure is illustrated in Figure 3, and the pseudo-code is provided in Algorithm 1.

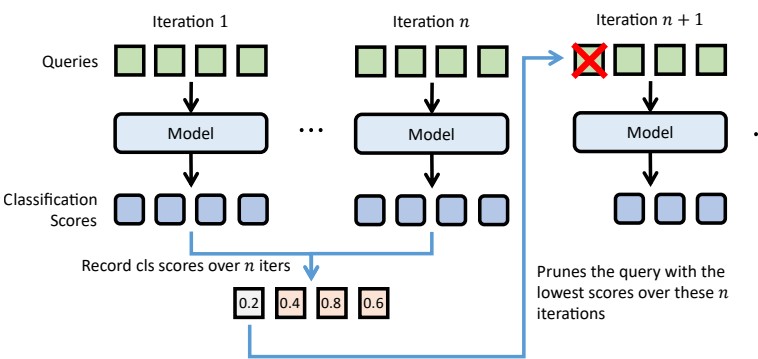

Figure 3: The pruning process. We select the query that generates the lowest classification scores every $n$ iterations. The selected query is then removed from the model and will no longer participate in any operations after being pruned.

---

**Algorithm 1: Gradually Pruning Queries**

---

**Input:** Total iterations $T$, the number of initial (final) queries $N_q$ ($N_q'$), a 3D detection model with queries $Q$, and pruning interval $n$.

1  Load the model from an existing checkpoint with queries $Q$
2  Initialize current iteration $t = 1$, and the number of current queries $N = N_q$
3  **while** $t \leq T$ **do**
4      Load images $I$, and extract image features $F$ from $I$
5      Update $Q$ through Transformer-Layers
6      Get classification scores $\mathcal{C}$ through the Classification-Branch
7      **if** $N > N_q'$ **then**
8          Record classification scores of each query
9      **if** $t \mod n = 0$ *and* $N > N_q'$ **then**
10         **Select the query that generates the lowest classification score relying on records**
11         **Remove the selected query from $Q$**
12         Update current query number $N \leftarrow N - 1$
13     $t \leftarrow t + 1$

**Output:** Final pruned queries $\mathcal{Q} \leftarrow Q$

---

### 3.3 WHY IS PRUNING QUERIES EFFECTIVE?

The reason why our method works is that queries are much independent between each others. Removing a certain query has slight impact to other queries. According to the rule of matrix multiplication, multiplying a matrix $A \in \mathbb{R}^{N_A \times M}$ by a matrix $B \in \mathbb{R}^{M \times N_B}$ is equivalent to multiplying each row $A_i \in \mathbb{R}^M$ of $A$ by $B$ individually, and then concatenating the results:

$$AB \equiv \underset{i=1,\cdots,N_A}{\text{Concat}} (A_i B) \tag{5}$$

where $N_A$, $N_B$ and $M$ are positive integers. If we delete the $i$-th row from $A$, the results of the multiplication involving the remaining rows with $B$ will remain unchanged.

In each MLP and cross-attention module, the query matrix $Q$ appears only once. This means that, according to Equ. 5, if we remove $Q_i$ from $Q$, the results of the other queries in the MLP and cross-attention modules will remain unaffected.

The only influence occurs in the self-attention modules. In the self-attention mechanism, the *query* $Q$ also serves as both the *key* $K$ and the *value* $V$. When a query $Q_i$ is removed, the other queries are affected because the right side of the matrix multiplication has also changed. According to Equ. 1, self-attention mechanism can be formated as:

$$\text{SelfAttention} = \text{Softmax} \left( \frac{(QW^Q)(QW^K)^{\text{T}}}{\sqrt{E}} \right) (QW^V) \cdot W^O \tag{6}$$

In multi-layer transformers, the queries interact with image features in the cross-attention module, so queries in the deeper layers of the transformer also contain feature information related to the input image during self-attention. At this stage, self-attention can partially replicate the function of

cross-attention by sampling image features. However, this sampling is indirect and has less impact compared to cross-attention. Therefore, we can remove these queries.

### 3.4 WHY NOT TRAIN A NEW MODEL USING ONLY A FEW QUERIES?

We introduce a query pruning method to reduce the computational load for DETR-based detectors. A natural question arises: why not train the model with fewer queries from the beginning? On one hand, as discussed in Section 1, training with a larger number of queries enhances the model's capacity to adapt to a diverse range of objects. By using GPQ, it becomes possible to prune a checkpoint trained with many queries, producing models with different query counts. This approach offers flexibility for various scenarios without necessitating retraining for each specific query configuration.

On the other hand, a key advantage of GPQ is its ability to retain the knowledge gained during training with a higher query count. This leads to more accurate predictions compared to training a model from scratch with fewer queries. As illustrated in Figure 4, the distinction between pruning redundant queries and training with fewer queries becomes more apparent. When redundant queries are pruned, the remaining queries cluster together and continue to occupy the original solution plane. In contrast, training with fewer queries from the outset results in a more scattered distribution of queries, reducing their overall representational effectiveness.

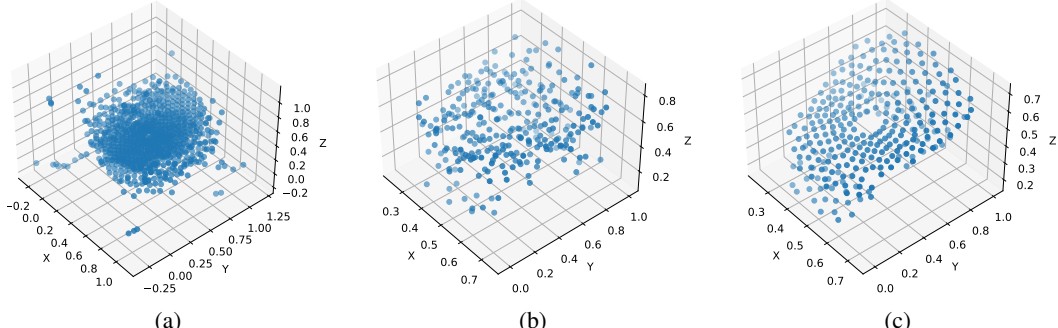

(a)  (b)  (c)

Figure 4: Illustration of reference points that are used to generate queries in PETRv2. (a) training using 900 queries; (b) training using 300 queries; (c) pruning from 900 to 300 queries. Training from scratch with fewer queries results in a more scattered and disordered distribution of queries.

## 4 EXPERIMENTS

### 4.1 EXPERIMENTAL SETUP

**Dataset and Detectors.** We conduct our experiments on the nuScenes dataset (Caesar et al., 2020), which consists of over 23,000 samples. The 3D detection task involves 10 object classes, including both static and dynamic objects.

To validate the effectiveness and efficiency of our proposed method, we perform experiments on five advanced detectors: DETR (Wang et al., 2022), PETR (Liu et al., 2022), PETRv2 (Liu et al., 2023a), FocalPETR (Wang et al., 2023a) and StreamPETR (Wang et al., 2023b). Notably, PETRv2 employs VovNet (Lee & Park, 2020) as its image backbone, while the remaining detectors utilize ResNet50 (He et al., 2016) as their image backbone.

**Evaluation Metrics.** In the field of 3D object detection, the primary performance evaluation metrics are mean Average Precision (mAP). NuScenes also provides the nuScenes Detection Score (NDS), which is derived from mAP, along with several other error metrics: mean Average Translation Error (mATE), mean Average Scale Error (mASE), mean Average Orientation Error (mAOE), mean Average Velocity Error (mAVE), and mean Average Attribute Error (mAAE). Mathematically, NDS $= \frac{1}{5}\left[5\text{mAP} + \sum_{\text{mTP}}(1 - \min(1, \text{mTP}))\right]$, where mTP $\in \{\text{mATE, mASE, mAOE, mAVE, mAAE}\}$.

Additionally, to assess the improvement in model runtime achieved by our method, we calculate the FLOPs (Floating Point Operations) for each module and record the model's runtime before and after pruning on resource-constrained edge devices. Typically, we use GFLOPs, where 1 GFLOPs $= 1 \times 10^9$ FLOPs.

## 4.2 Implementation Details

We use AdamW (Loshchilov & Hutter, 2019) optimizer with a weight decay of $1.0 \times 10^{-2}$. When training from scratch, we use a base learning rate of $2 \times 10^{-4}$ with a batch size of 8, and apply a cosine annealing policy (Loshchilov & Hutter, 2022) for learning rate decay. When fine-tuning from an existing checkpoint, the learning rate is set to $1.0 \times 10^{-4}$. The point cloud range is configured to $[-61.2\text{m}, 61.2\text{m}]$ along the $X$ and $Y$ axis, and $[-10.0\text{m}, 10.0\text{m}]$ along the $Z$ axis for all methods.

For all methods, classification and bounding box losses are applied with respective loss weights of 2.0 and 0.25. In line with the original publications, an additional 2D auxiliary task is introduced for both FocalPETR and StreamPETR. Furthermore, PETRv2 and StreamPETR leverage temporal information by incorporating historical frames into their models. During inference, the top 300 predictions with the highest classification scores are selected as the final results.

## 4.3 Main Results

### 4.3.1 Comparison of Detection Performance Before and After Pruning

| Model | Backbone | Image Size | Queries | FPS↑ | mAP↑ | NDS↑ | mATE↓ | mASE↓ | mAOE↓ | mAVE↓ | mAAE↓ |
|---|---|---|---|---|---|---|---|---|---|---|---|
| DETR3D | ResNet50 | 1408x512 | 900 / - | 8.2 | 24.63% | 0.3054 | 0.9534 | 0.2867 | 0.7005 | 0.9959 | 0.2417 |
| | | | 300 / - | 9.2 | 23.43% | 0.2953 | 0.9851 | 0.2865 | 0.7189 | 0.9667 | 0.2613 |
| | | | 150 / - | 9.2 | 20.55% | 0.2694 | 1.0045 | 0.2925 | 0.7426 | 1.1308 | 0.2981 |
| | | | 900 / 300 | 9.3 | **24.78%** | **0.3234** | **0.9404** | **0.2789** | **0.6139** | **0.9397** | 0.2326 |
| | | | 900 / 150 | 9.4 | 22.63% | 0.3015 | 0.9713 | 0.2813 | 0.6444 | 0.9919 | **0.2276** |
| PETR | ResNet50 | 1408x512 | 900 / - | 7.1 | 31.74% | 0.3668 | 0.8395 | 0.2797 | 0.6153 | 0.9522 | 0.2322 |
| | | | 300 / - | 8.9 | 31.19% | 0.3536 | 0.8449 | 0.2872 | 0.6156 | 1.0673 | 0.2762 |
| | | | 150 / - | 9.2 | 28.37% | 0.3158 | 0.8664 | 0.2899 | 0.7340 | 1.1074 | 0.3706 |
| | | | 900 / 300 | 8.9 | **32.85%** | **0.3884** | **0.8003** | **0.2791** | **0.5507** | **0.9108** | **0.2179** |
| | | | 900 / 150 | 9.3 | 30.52% | 0.3671 | 0.8237 | 0.2792 | 0.5804 | 0.9441 | 0.2282 |
| | VovNet | 1600x640 | 900 / - | 3.1 | **40.45%** | **0.4517** | 0.7282 | **0.2706** | 0.4482 | **0.8404** | 0.2179 |
| | | | 300 / - | 3.8 | 38.98% | 0.4279 | 0.7636 | 0.2732 | 0.4820 | 0.9198 | 0.2315 |
| | | | 900 / 300 | 3.7 | 40.04% | 0.4507 | **0.7278** | 0.2723 | **0.4383** | 0.8451 | **0.2110** |
| PETRv2 | VovNet | 800x320 | 900 / - | 5.5 | **40.64%** | **0.4949** | **0.7374** | 0.2693 | 0.4636 | 0.4162 | 0.1967 |
| | | | 300 / - | 6.5 | 39.19% | 0.4893 | 0.7595 | **0.2678** | **0.4416** | 0.4360 | 0.1916 |
| | | | 150 / - | 6.8 | 38.00% | 0.4709 | 0.7710 | 0.2760 | 0.4773 | 0.4652 | 0.2013 |
| | | | 900 / 300 | 6.7 | 40.26% | 0.4944 | 0.7383 | 0.2701 | 0.4542 | **0.4146** | **0.1916** |
| | | | 900 / 150 | 6.9 | 39.16% | 0.4919 | 0.7385 | .02702 | 0.4271 | 0.4135 | 0.1898 |
| FocalPETR | ResNet50 | 704x256 | 900 / - | 16.4 | 32.44% | 0.3752 | 0.7458 | **0.2778** | 0.6489 | 0.9458 | 0.2522 |
| | | | 300 / - | 19.3 | 31.59% | 0.3524 | 0.7594 | 0.2838 | 0.7154 | 1.0432 | 0.2973 |
| | | | 150 / - | 21.2 | 27.78% | 0.3071 | 0.8276 | 0.2826 | 0.7863 | 1.2156 | 0.4178 |
| | | | 900 / 300 | 19.6 | **33.17%** | **0.3925** | **0.7446** | 0.2800 | **0.6265** | **0.8619** | **0.2203** |
| | | | 900 / 150 | 21.2 | 31.81% | 0.3834 | 0.7563 | 0.2829 | 0.6119 | 0.8792 | 0.2259 |
| StreamPETR | ResNet50 | 704x256 | 900 / - | 16.1 | 37.83% | 0.4734 | 0.6961 | 0.2822 | 0.6846 | 0.2856 | 0.2084 |
| | | | 300 / - | 18.5 | 33.62% | 0.4429 | 0.7305 | 0.2837 | 0.6800 | 0.3333 | 0.2251 |
| | | | 150 / - | 18.8 | 26.46% | 0.3763 | 0.8195 | 0.2921 | 0.8135 | 0.3998 | 0.2353 |
| | | | 900 / 300 | 18.7 | **39.42%** | **0.4941** | **0.6766** | **0.2711** | **0.5799** | **0.2708** | 0.2136 |
| | | | 900 / 150 | 19.3 | 34.94% | 0.4633 | 0.6989 | 0.2749 | 0.6226 | 0.3124 | **0.2050** |

Table 2: Pruning results for different models. The column of "Queries" is of format "# Initial queries/ # Final queries". Lines where "# Final queries" remain blank are baselines used to compare. All baselines and checkpoints are trained for 24 epochs, with the pruning process completed within the first 6 epochs when loading a checkpoint. The FPS is measured on a single RTX3090 GPU.

One of the key objectives of our pruning method is to maintain the performance of the original models. As demonstrated in Table 2, our approach successfully preserves, and in some cases enhances, the performance of various detectors, even when pruning a checkpoint with a large number of queries. GPQ can also accelerate model inference on desktop GPUs. For example, in the case of PETR, pruning from 900 to 150 queries results in an mAP of 30.52%, which is 2.15 points higher than training from scratch with 150 queries (28.37%), and its speed increases from 7.1 fps to 9.3, which is 1.31x faster. Remarkably, pruning from 900 to 300 queries achieves an mAP of 32.85%, outperforming the result of training from scratch with 900 queries (31.74%). Similarly, for both FocalPETR and StreamPETR, pruning from 900 to 300 queries yields optimal performance, surpassing the results obtained from training with 900 queries from scratch. For PETRv2, while pruning from

900 to 300 queries results in a slightly lower mAP than training from scratch with 900 queries, it still exceeds the performance of training from scratch with 300 queries.

For 3D objects, their poses and moving states are also important. the NDS of pruned models with a final query count of 300 is higher than that of models with 900 queries for PETR, FocalPETR, and StreamPETR. In the case of PETRv2, although the NDS slightly decreases for pruned models, it remains comparable to the performance with 900 queries.

In summary, the experimental results show that our pruning method significantly reduces the number of queries while maintaining, or even slightly improving, performance compared to models trained from scratch with a larger number of queries. Additionally, the pruned models consistently outperform those trained from scratch with the same number of queries.

### 4.3.2 COMPARISON OF INFERENCE SPEED BEFORE AND AFTER PRUNING

| Model | Backbone | Pruned | # Queries | GFLOPs | Reduced FLOPs | Time (ms) | Reduced Time |
|-------|----------|--------|-----------|--------|---------------|-----------|--------------|
| PETR | ResNet18 | ✗ | 900 | 219.14 | - | 1829.44 | - |
| | | ✓ | 300 | 164.61 | 24.89% | 1231.46 | 32.69% |
| | | ✓ | 150 | 152.06 | 30.61% | 1103.96 | 39.66% |
| | w/o | ✗ | 900 | 99.39 | - | 1140.97 | - |
| | | ✓ | 300 | 44.86 | 54.87% | 563.85 | 50.58% |
| | | ✓ | 150 | 32.30 | 67.50% | 439.34 | 61.49% |
| FocalPETR | ResNet18 | ✗ | 900 | 162.36 | - | 1319.35 | - |
| | | ✓ | 300 | 118.07 | 27.28% | 846.62 | 35.83% |
| | | ✓ | 150 | 108.08 | 33.44% | 745.44 | 43.50% |
| | w/o | ✗ | 900 | 78.07 | - | 868.28 | - |
| | | ✓ | 300 | 33.77 | 56.75% | 416.12 | 52.07% |
| | | ✓ | 150 | 23.77 | 69.55% | 319.84 | 75.76% |
| StreamPETR | ResNet18 | ✗ | 900 | 172.08 | - | 1520.07 | - |
| | | ✓ | 300 | 123.90 | 28.00% | 916.03 | 39.74% |
| | | ✓ | 150 | 112.51 | 34.62% | 791.08 | 47.96% |
| | w/o | ✗ | 900 | 87.78 | - | 1030.38 | - |
| | | ✓ | 300 | 39.59 | 54.90% | 477.81 | 53.63% |
| | | ✓ | 150 | 28.21 | 67.86% | 359.00 | 76.38% |

Table 3: Running time on the Jetson Nano B01 device with and without backbone. Data preparation time is excluded from the measurements. FlashAttention is not used, as it is not supported by ONNX. Randomly generated dummy input images are used for testing, with sizes of $704 \times 256$ for FocalPETR and StreamPETR, and $800 \times 320$ for PETR.

Inference speed is crucial for deploying models on edge devices. To verify whether pruning queries can indeed improve speed, we export the model to ONNX format and deploy it on the Jetson Nano B01 using ONNX Runtime to measure the model's running time. After pruning from 900 to 300 queries, the FLOPs of StreamPETR are reduced by 28%, and the running time decreases by 39.74%. Pruning further from 900 to 150 queries results in a 47.96% reduction in running time, making the model $1.92\times$ faster. The pruning method also proves effective for FocalPETR and PETR, significantly reducing both FLOPs and running time. Further details can be found in Table 3.

Since our method does not modify the image backbones or necks, we remove the backbone module to better illustrate the effects of our approach. Specifically, we run only the transformer decoder using randomly generated dummy inputs. Results are also shown in Table 3. For all the evaluated models, pruning from 900 queries to 300 results in saving more than half of the running time for transformer-related modules. Additionally, compared to the model with 900 queries, StreamPETR with 150 pruned queries achieves a 76.38% reduction in running time.

### 4.3.3 COMPARISON WITH ToMe

To our knowledge, this is the first study to explore query redundancy in Transformer-based 3D detection models. As a result, it is challenging to find comparable methods for direct evaluation. However, in this section, we compare our approach with ToMe (Bolya et al.), a method that shares a similar goal of improving efficiency. Specifically, ToMe increases the throughput of Vision Transformer

(ViT) models by dividing $N$ tokens into two equal size groups $\mathbb{A}$ and $\mathbb{B}$. These tokens represent image features extracted by the image backbone. To merge similar tokens, ToMe calculates a similarity matrix $\mathbf{S}$ of size $\frac{N}{2} \times \frac{N}{2}$, where the $(i, j)$-th entry represents the similarity between token $i$ in group $\mathbb{A}$ and token $j$ in group $\mathbb{B}$. It then merges $r$ token pairs with the highest similarity scores.

We apply ToMe to StreamPETR, and compare its performance with GPQ in Table 4. Surprisingly, instead of accelerating computation, ToMe causes the model to run slower. We believe this is due to the overhead introduced by calculating the similarity matrix. Unlike image classification tasks, which typically handle only a few hundred tokens, 3D object detection methods often generate a much larger number of tokens (e.g., StreamPETR generates 4224 tokens even with a relatively small image size of 704×256). This increased token size significantly amplifies the computational cost of constructing the similarity matrix, making ToMe inefficient for 3D object detection tasks. Compared to ToMe, GPQ not only preserves the model's performance but also achieves faster inference speed.

| Model | r | Queries | Speed(fps) | mAP ↑ | NDS ↑ | mATE ↓ | mASE ↓ | mAOE ↓ | mAVE ↓ | mAAE ↓ |
|---|---|---|---|---|---|---|---|---|---|---|
| StreamPETR | - | 900/- | 16.1 | 37.83% | 0.4734 | 0.6961 | 0.2822 | 0.6846 | 0.2856 | **0.2084** |
| StreamPETR-ToMe | 264 | 900/- | 14.8 | 37.79% | 0.4731 | 0.6982 | 0.2819 | 0.6849 | 0.2849 | 0.2084 |
| StreamPETR-ToMe | 528 | 900/- | 14.9 | 37.69% | 0.4721 | 0.6994 | 0.2822 | 0.6855 | 0.2877 | 0.2088 |
| StreamPETR-ToMe | 1056 | 900/- | 15.4 | 36.34% | 0.4608 | 0.7178 | 0.2852 | 0.6963 | 0.2965 | 0.2121 |
| StreamPETR-ToMe | 2112 | 900/- | 16.0 | 31.69% | 0.4325 | 0.7546 | 0.2907 | 0.6893 | 0.3170 | 0.2210 |
| StreamPETR-GPQ | - | 900/300 | **18.7** | **39.42%** | **0.4941** | **0.6766** | **0.2711** | **0.5799** | **0.2780** | 0.2136 |

Table 4: Comparison with ToMe. Results indicate that ToMe performs poorly on 3D object detection, while our GPQ effectively maintains the model's performance and accelerates inference.

## 4.4 ABLATION EXPERIMENTS

| Model | mAP ↑ | NDS ↑ | mATE ↓ | mASE ↓ | mAOE ↓ | mAVE ↓ | mAAE ↓ |
|---|---|---|---|---|---|---|---|
| StreamPETR | 37.83% | 0.4734 | 0.6961 | 0.2822 | 0.6846 | 0.2856 | **0.2084** |
| prune highest | 34.34% | 0.4563 | 0.7429 | 0.2813 | 0.5912 | 0.3188 | 0.2195 |
| prune using cost | 38.78% | 0.4899 | 0.6808 | 0.2814 | 0.5791 | 0.2853 | 0.2130 |
| prune in one iteration | 35.71% | 0.4677 | 0.7121 | 0.2828 | 0.5970 | 0.2930 | 0.2233 |
| GPQ | **39.42%** | **0.4941** | **0.6766** | **0.2711** | **0.5799** | **0.2780** | 0.2136 |

Table 5: Ablation experiments. We use StreamPETR as baseline, and all pruning strategies start from a checkpoint initialized with 900 queries and reduce the number to 300 through pruning.

**Pruning Criterion.** Our pruning strategy removes the queries with the lowest classification scores. To validate this criterion, we conducted two comparative experiments: one where we prune queries with the highest classification scores (**prune highest** in Table 5) and another where pruning is guided by the cost produced by the assigner during the binary matching process between predicted and ground truth values (**prune using cost** in Table 5).

Table 5 shows that pruning queries with the highest classification scores leads to a noticeable performance drop compared to our default strategy. While using the cost generated by the assigner as the pruning criterion results in performance closer to the original model, it still falls short of the performance achieved by GPQ. These results confirm the effectiveness of our proposed method.

**Pruning Strategy.** A key feature of our method is the gradual pruning strategy. To validate its effectiveness, we conducted an experiment where 600 queries were pruned in a single iteration (**prune in one iteration** in Table 5). The results show a significant performance drop when all queries are pruned in a single step instead of gradually. This demonstrates that the gradual pruning strategy employed in GPQ is not only reasonable but also the optimal approach for query pruning.

## 5 CONCLUSION

In this paper, we propose GPQ, an incredibly simple yet effective pruning method that gradually eliminates redundant queries in DETR-based 3D detection models, aiming to reduce computational cost without compromising performance. Results on the nuScenes dataset confirm that our method effectively maintains the detection performance across all evaluated models. To the best of our knowledge, this is the first study to explore query pruning in query-based models, and we hope our work will inspire further research into pruning DETR-based models.

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

# A   MORE EXPERIMENTAL RESULTS

In the appendix, we provide more experimental results of GPQ, including:

- Qualitative results of GPQ (A.1).
- Extended experimental results of GPQ on 2D object detection (A.2).
- Further experimental results combining GPQ with training from scratch (A.3).
- More experimental results for fully converged models (A.4).

## A.1   VISUALIZATION RESULTS

To further validate the effectiveness of our method, we visualize the results of evaluated 3D models used in the experiments before and after pruning, as shown in Figure 5. As demonstrated, pruning does not negatively impact the detection performance for both static and moving objects, further confirming the robustness of our pruning approach.

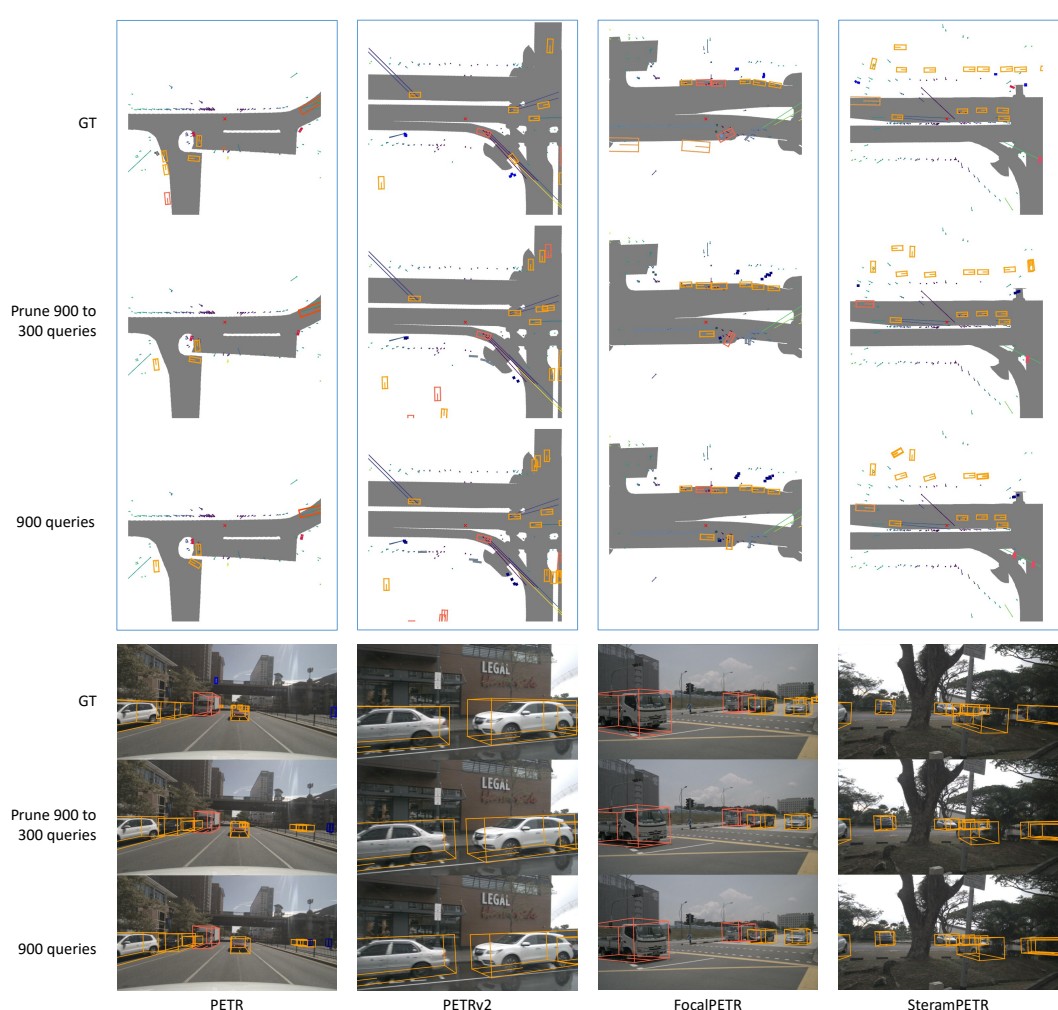

Figure 5: Visualization results of the evaluated models from a top-down view (top) and from camera perspectives (bottom) before and after pruning. Through comparison, we can further confirm that our method effectively preserves the models' performance.

## A.2   RESULTS ON 2D OBJECT DETECTION

3D object detection is a highly practical research area with fewer conversion steps required for its application in real-world commercial scenarios. In addition, as discussed in Section 1, the queries

in 3D object detection models exhibit a high degree of redundancy. In contrast, 2D object detection methods typically use 300 queriesCarion et al. (2020); Meng et al. (2021) to predict 80 classes (e.g., COCO dataset(Lin et al., 2014)), resulting in a lower level of query redundancy. This is why we chose to focus on 3D object detection from the beginning, rather than 2D object detection or 2D image classification.

However, because pre-defined queries also exist in DETR, we can still apply GPQ to DETR-based 2D object detection methods. We conducted experiments using ConditionalDETR as an example.

| Model | Backbone | Queries | FPS | mAP | $AP_{50}$ | $AP_{75}$ | $AP_s$ | $AP_m$ | $AP_l$ |
|---|---|---|---|---|---|---|---|---|---|
| ConditionalDETR | ResNet50 | 300/- | 18.6 | 0.409 | 0.618 | 0.434 | 0.206 | 0.442 | 0.591 |
| | | 150/- | 23.3 | 0.398 | 0.606 | 0.421 | 0.196 | 0.432 | 0.582 |
| | | 300/150 | 23.5 | 0.406 | 0.615 | 0.429 | 0.197 | 0.439 | 0.598 |

Table 6: Results of GPQ applied to ConditionalDETR. The original ConditionalDETR uses 300 queries with 50 epochs. We prune half of these in 2 epochs and then fine-tune for 6 epochs, which costs much less than training a new model using 150 queries.

As shown in Table 6, after pruning half of the queries using GPQ, ConditionalDETR is still able to maintain its original performance. Moreover, the pruning process requires only 8 epochs, which is significantly more efficient compared to retraining a new model from scratch (50 epochs). These results demonstrate that our method can still achieve its intended effect when applied to 2D object detection methods.

## A.3 INTEGRATING WITH TRAINING

A question may arise: for previous models with existing checkpoints, we can directly use GPQ to prune an existing checkpoint. But for future methods, does GPQ still have its value? If we train a model first and then prune it, why not train with fewer queries for more epochs?

Fortunately, GPQ can be integrated with training. As show in Table 7, we use PETR(Liu et al., 2022) and StreamPETR(Wang et al., 2023b) as examples, starting with 900 queries, and gradually pruning during training, ultimately reaching a state of 300 queries. With this technique, one can directly train a model using GPQ without the need for an additional pruning and fine-tuning stage or more epochs to achieve the same performance with fewer queries.

| Model | mAP ↑ | NDS ↑ | mATE ↓ | mASE ↓ | mAOE ↓ | mAVE ↓ | mAAE ↓ |
|---|---|---|---|---|---|---|---|
| PETR-900q | 31.74% | **0.3668** | 0.8395 | **0.2797** | 0.6153 | **0.9522** | **0.2322** |
| PETR-300q | 31.19% | 0.3536 | 0.8449 | 0.2872 | 0.6156 | 1.0673 | 0.2762 |
| PETR-GPQ-t | **31.75%** | 0.3644 | **0.8336** | 0.2802 | **0.6028** | 0.9878 | 0.2393 |
| StreamPETR-900q | **37.83%** | **0.4734** | 0.6961 | 0.2822 | 0.6846 | **0.2856** | **0.2084** |
| StreamPETR-300q | 33.62% | 0.4429 | 0.7305 | 0.2837 | 0.6800 | 0.3333 | 0.2251 |
| StreamPETR-GPQ-t | 36.41% | 0.4673 | **0.6948** | **0.2806** | **0.6423** | 0.3079 | 0.2221 |

Table 7: Results of integrating GPQ with training. All models use ResNet50 as backbone, with an image size of $1408 \times 512$. Here, PETR-GPQ-t and StreamPETR-GPQ-t start without loading a pre-trained PETR checkpoint, begin with 900 queries, and are pruned gradually to a final state of 300 queries.

## A.4 FULLY CONVERGED MODELS

To evaluate the performance of GPQ on fully converged models, we had to increase the number of training steps. However, due to limited computational resources, it is not feasible to indefinitely increase training time and epochs. Through experimentation, we are confident that by 90 epochs, StreamPETR has either fully converged or is close to full convergence(Table 8). Therefore, we performed pruning on the checkpoint trained for 90 epochs. As shown in Table 8, applying GPQ to prune a fully converged 900-query model down to 300 queries resulted in better performance than

the fully converged 300-query model. This further demonstrates that GPQ can still be effective even after the model has fully converged.

| Model | mAP ↑ | NDS ↑ | mATE ↓ | mASE ↓ | mAOE ↓ | mAVE ↓ | mAAE ↓ |
|---|---|---|---|---|---|---|---|
| StreamPETR-300q-24e | 33.62% | 0.4429 | 0.7305 | 0.2837 | 0.6800 | 0.3333 | 0.2251 |
| StreamPETR-300q-36e | 38.65% | 0.4890 | 0.6885 | 0.2769 | 0.5871 | 0.2823 | 0.2081 |
| StreamPETR-300q-48e | 39.72% | 0.4978 | 0.6782 | 0.2762 | 0.5662 | 0.2866 | 0.2010 |
| StreamPETR-300q-60e | 41.83% | 0.5194 | 0.6399 | 0.2764 | 0.5099 | 0.2678 | 0.2031 |
| StreamPETR-300q-90e | 42.00% | 0.5280 | 0.6224 | 0.2717 | 0.4464 | 0.2703 | 0.2091 |
| StreamPETR-900q-90e | 43.23% | 0.5369 | 0.6093 | 0.2701 | 0.4449 | 0.2791 | 0.1893 |
| StreamPETR-900q-GPQ | 42.49% | 0.5301 | 0.6237 | 0.2709 | 04557 | 0.2799 | 0.1928 |

Table 8: Results of GPQ on fully converged models. Here 300q-24e means the model was trained with 300 queries for 24 epochs. StreamPETR-900q-GPQ loaded the checkpoint of StreamPETR-900q-90e and pruned it into 300 queries.

