# OpenReview forum: "Redundant Queries in DETR-Based 3D Detection Methods: Unnecessary and Prunable"
_ICLR.cc/2025/Conference — ICLR 2025 Conference Withdrawn Submission_

### Official Review · Reviewer_KmgM · 2024-11-02

**Soundness:** 2
**Presentation:** 3
**Contribution:** 2
**Rating:** 6
**Confidence:** 3

**Summary:**

This paper addresses the issue of redundancy in DETR-based 3D object detection models. The authors find that not all queries contribute equally to the detection task, and a significant portion of them have a minimal impact. They propose an approach called Gradually Prunes Queries (GPQ), which reduces the number of queries by pruning those with lower classification scores. The overall framework appears simple. However, my main concern lies in the discussion of related work. There is indeed a substantial amount of work aimed at speeding up query-based methods in 2D object detection. The authors did not provide a convincing and comprehensive review, nor did they offer a comparison with the most relevant methods.

**Strengths:**

1. The paper is well-structured, with a clear abstract, introduction, methodology, experiments, and conclusion sections that logically flow from one to the next.
2. The proposed Gradually Prunes Queries method is simple and effective.
3. The experiment results, when compared to different standard detectors, show that the design is effective.

**Weaknesses:**

1. There are some works aimed at speeding up query-based methods in 2D object detection, but this paper did not discuss. So, it is not clear if very similar methods exist in 2D object detection.
[1] Yang, Chenhongyi, Zehao Huang, and Naiyan Wang. "QueryDet: Cascaded sparse query for accelerating high-resolution small object detection." Proceedings of the IEEE/CVF Conference on computer vision and pattern recognition. 2022.
[2] Zhu, Yuan, Qingyuan Xia, and Wen Jin. "Srdd: a lightweight end-to-end object detection with transformer." Connection Science 34.1 (2022): 2448-2465.
2. It is not clear what the special challenges are in 3D space, and why the experiments are only conducted in 3D space.
3. Using fewer queries may lead to slower network convergence, so fewer queries might require more training iterations. However, since the same number of epochs were used in Table 1, it's difficult to conclude that the precision drops quickly solely due to fewer queries.

**Questions:**

1.	Are there very similar methods proposed in the field of 2D object detection?
2.	Has there been a comparison with other methods that reduce the number of queries or speed up DETR? What challenges are there in transferring these methods to 3D object detection? How about the tested efficiency?

---

> ### Author Response · Authors · 2024-11-23
>
> We appreciate your feedback and will proceed to address your concerns in the order they appear.
>
> # Response to Weakness
>
> 1. We have also noted the concerns regarding similar works.
>     - QueryDet has little relation to this work and shares low similarity. It is not transformer-based, and its queries are not closely related to those in transformer methods. Furthermore, its approach is unrelated to model pruning.
>     - The SRDD method focuses on tokens in the transformer encoder, whereas the DETR-like methods targeted in this work typically consist of only a transformer decoder structure. SRDD is more aligned with traditional model pruning approaches and has no connection to query pruning.
>     - To date, we have not identified any methods similar to ours.
>
> 2. Your concerns are reasonable. We will explain why we focus on the 3D object detection field. **3D object detection is a highly practical research area** with fewer conversion steps required for its application in real-world commercial scenarios. This is why we chose to focus on 3D object detection from the beginning, rather than 2D object detection or 2D image classification. Furthermore, research on model pruning for 3D object detection methods is currently limited and existing pruning methods for ViT models and 2D object detection models are difficult to be applied for 3D object detection methods. We acknowledge that the discussion on this matter in the paper was insufficient. Thank you for pointing that out. We have added this content in the updated version of the paper, Section 2.
>
>   However, because pre-defined queries also exist in DETR, we can still apply GPQ to DETR-based 2D object detection methods. We conducted experiments using Conditional DETR as an example, and the results are shown in the table below. As shown in the table, after pruning half of the queries using GPQ, Conditional DETR is still able to maintain its original performance. Moreover, the pruning process requires only 8 epochs, which is significantly more efficient compared to retraining a new model from scratch (50 epochs). These results demonstrate that our method can still achieve its intended effect when applied to 2D object detection methods.
>
>
> |     Model      | Backbone | Queries | FPS | FPSx |   mAP | $\mathbf{AP}_{50}$ | $\mathbf{AP}_{75}$ | $\mathbf{AP}_s$ | $\mathbf{AP}_m$ | $\mathbf{AP}_l$ |
> | :------------: | :------: | :------ | ----: | ---------------: | ---------------: | ------------: | ------------- | ------------: | ------------- | ------------: |
> | ConditionaDETR | ResNet50 | 300/-   | 18.6 | 1.00x | 0.409 |            0.618 |            0.434 |         0.206 | 0.442         |         0.591 |
> | ConditionaDETR | ResNet50 | 150/-   | 23.3 | 1.25x | 0.398 | 0.606 | 0.421 | 0.196 | 0.432 | 0.582 |
> | ConditionaDETR | ResNet50 | 300/150 | 23.5 | 1.26x | 0.406 |            0.615 |            0.429 |         0.197 | 0.439         |         0.598 |
>
>
> 3. Higher iteration steps generally lead to higher model accuracy but also result in longer training times. Theoretically, assuming no overfitting, infinitely long training could bring accuracy infinitely close to 1. To enable fair model comparisons, accuracy is typically evaluated with a fixed number of epochs.
>    While models with fewer queries may eventually catch up to the accuracy of models with more queries as training iterations increase, this would come at the cost of longer training times. Therefore, following community best practices, we compare the accuracy of models with different numbers of queries under the same number of training steps.
>
> # Response to Questions
>
> 1. To date, no methods specifically focused on pruning queries have been identified, and there is no "very similar" work in this area.
>
> 2. To date, pruning queries in DETR-based models remains an unexplored area. To the best of our knowledge, GPQ is the first work to explore pruning queries in DETR-based methods.
>   However, there are still many methods that tries to prune tokens in ViT models. As we supplemented in Section 2 of updated paper, they are difficult to be migrated to 3D object detection models due to following reasons:
>     1. Nonexistent Pruning Targets. Some pruning targets in ViT models do not exist in 3D object detection models.
>     2. Structural Inconsistencies. Some methods rely on specific structures that do not exist in 3D object detection.
>     3. Massive Data Differences. 3D object detection models generate larger feature maps, making it challenging to transfer existing methods.

---

> > ### Comment · Reviewer_KmgM · 2024-11-24
> > **Response to Author**
> >
> > Thanks for the author's response, the author solved my first two concerns. However, regarding the third concern, I still have question. The reason is that the method in this paper needs to be initialized with a pre-trained model, which means that the method in this paper uses a longer training time. So I am eager to know, is there a way to achieve the results of this paper by simply increasing the training time with a small number of queries?

---

> > > ### Author Response · Authors · 2024-11-25
> > >
> > > Thank you for your comment!
> > >
> > > Yes, using fewer queries while increasing the number of training epochs can indeed lead to better performance. Taking StreamPETR with 300 queries as an example, the results are shown in the table below.
> > >
> > > Epochs|mAP|NDS|mATE|mASE|mAOE|mAVE|mAAE
> > > ---|---|---|---|---|---|---|---
> > > 24|33.62%|0.4429|0.7305|0.2837|0.6800|0.3333|0.2251
> > > 36|38.65%|0.4890|0.6885|0.2769|0.5871|0.2823|0.2081
> > > 48|39.72%|0.4978|0.6782|0.2762|0.5662|0.2866|0.2010
> > >
> > >
> > > However, the goal of our method is not to retrain a model and then prune it. As stated in the abstract of our paper, many existing methods provide numerous checkpoints, and our approach aims to improve the runtime efficiency of these models during deployment by pruning queries after obtaining a model's checkpoint. Therefore, our method does not require training a new model from scratch.
> > >
> > > In our experiments, we retrained many models for the sake of fair comparisons. However, in practical applications, we can directly start with an existing checkpoint and prune it to obtain a smaller model. For example, we use the StreamPETR checkpoint provided by the official repo, which was trained with ResNet50 as the backbone for 90 epochs, and pruned it to 300 queries. The results are shown in the table below. When pruning, we can freeze the backbone, only fine-tune the transformer-decoder. This process incurs significantly lower costs than retraining a 300-query model for 90 epochs.
> > >
> > > Model | mAP | NDS | mATE | mASE | mAOE | mAVE | mAAE
> > > ---|---|---|---|---|---|---|---
> > > 24e|37.83%|0.4734|0.6961|0.2822|0.6846|0.2856|0.2084
> > > 24e-GPQ|39.42%|0.4941|0.6766|0.2711|0.5799|0.2708|0.2136
> > > 36e|41.34%|0.5110|0.6538|0.2744|0.5580|0.2710|0.1999
> > > 36e-GPQ|41.00%|0.5144|0.6410|0.2750|0.5050|0.2833|0.2022
> > > 90e|43.23%|0.5369|0.6093|0.2701|0.4449|0.2791|0.1893
> > > 90e-GPQ|42.49%|0.5301|0.6237|0.2709|0.4557|0.2799|0.1928
> > >
> > > Surprisingly, we found that our method can be integrated with the training process. This means that for future methods, GPQ can be applied during training by initially setting a larger number of queries. The final model will have fewer queries without introducing any additional training overhead. Results are shown in table below.
> > >
> > > Model | mAP | NDS | mATE | mASE | mAOE | mAVE | mAAE
> > > ---|---|---|---|---|---|---|---
> > > PETR-900q|31.74%|0.3668|0.8395|0.2797|0.6153|0.9522|0.2322
> > > PETR-300q|31.19%|0.3536|0.8449|0.2872|0.6156|1.0673|0.2762
> > > PETR-GPQ-t|31.75%|0.3644|0.8336|0.2802|0.6028|0.9878|0.2393
> > >
> > > Here, `PETR-GPQ-t` starts without loading a pre-trained PETR checkpoint, begins with 900 queries, and prunes gradually to a final state of 300 queries.
> > > This result further confirms the versatility of our method.

---

> > > > ### Comment · Reviewer_KmgM · 2024-11-26
> > > > **Response to Author**
> > > >
> > > > Thanks for the author's honest response. Now, I think GPQ  can indeed save some training time and improve some reasoning efficiency.  But considering that the method works only on no fully converged checkpoints, but not on checkpoints converged well, I raise my score to borderline accept.

---

> > > > > ### Author Response · Authors · 2024-11-27
> > > > >
> > > > > Thank you very much for your positive feedback!
> > > > >
> > > > > After understanding your latest concerns, we conducted further experiments. To ensure the model fully converges, more training steps are required. Honestly, this demands significant computational resources. We ran the StreamPETR 300-query model for 90 epochs, which is the maximum number of epochs we could afford. As shown in the table below, with more epochs, the improvement in mAP becomes slower. We have reason to believe that by 90 epochs, StreamPETR has fully converged.
> > > > >
> > > > > Based on this, we obtained fully converged checkpoints for both 900-query and 300-query models, and then applied GPQ pruning to the 900-query checkpoint. Here `StreamPETR-GPQ` loads the checkpoint of `StreamPETR-900q-90e` and prunes it into 300 queries. The result showed that, using our method, the performance was better than the 300-query model.
> > > > >
> > > > > We hope our results help address your concerns!
> > > > >
> > > > > Model        | mAP $\uparrow$ | NDS $\uparrow$ | mATE $\downarrow$ | mASE $\downarrow$ | mAOE $\downarrow$ | mAVE $\downarrow$ | mAAE $\downarrow$
> > > > > ---------------------|-------|-------|--------|--------|--------|--------|--------
> > > > > StreamPETR-300q-24e|33.62%|0.4429|0.7305|0.2837|0.6800|0.3333|0.2251
> > > > > StreamPETR-300q-36e|38.65%|0.4890|0.6885|0.2769|0.5871|0.2823|0.2081
> > > > > StreamPETR-300q-48e|39.72%|0.4978|0.6782|0.2762|0.5662|0.2866|0.2010
> > > > > StreamPETR-300q-60e|41.83%|0.5194|0.6399|0.2764|0.5099|0.2678|0.2031
> > > > > StreamPETR-300q-90e|42.00%|0.5280|0.6224|0.2717|0.4464|0.2703|0.2091
> > > > > StreamPETR-900q-90e|43.23%|0.5369|0.6093|0.2701|0.4449|0.2791|0.1893
> > > > > StreamPETR-GPQ  |42.49%|0.5301|0.6237|0.2709|0.4557|0.2799|0.1928

---

### Official Review · Reviewer_EdfF · 2024-11-03

**Soundness:** 2
**Presentation:** 3
**Contribution:** 2
**Rating:** 6
**Confidence:** 4

**Summary:**

The manuscript proposes to prune queries in DETR-based 3D detectors to speed up inference by almost 2x without loss of detection performance (in some cases even improving performance). Pruning is performed after training has finished without the need for any gradients to flow. The lowest scoring queries are incrementally removed one by one.

**Strengths:**

- The paper is well written and and illustrated. The figures help support the text i.e. Fig 1 and 3.
- The approach is well motivated through experiment: Fig 2 shows that the imbalanced number of selections per query is prevalent across various DETR-based detectors. There indeed seem to be some queries that are used very often and others that are not.
- The experiments are thorough for the proposed approach (but lack deeper/broader investigations see weaknesses). The approach is evaluated on four current PETR-based architectures and show across the board that the pruning increases throughput and that detection performance is maintained or slightly improved. The last point is positively surprising.
- The proposed approach is indeed straight forward and it is great that it can be deployed on existing checkpoints after training. This makes it universally applicable to 3D DETR models. Simple and universally applicable enables impact.

**Weaknesses:**

- The manuscript could have tried harder to investigate why training on a large set of queries and pruning them after is a better choice than training with fewer queries to begin with. The experiments show this (which is good), but the reader is left wondering about the why.
    - Which are the queries that get pruned? Is it that they attend to similar areas in space but always loose out against more confident queries?
    - If we took the top most confident queries (as per the proposed algorithm) and the least confident ones (invert the algorithm and prune the most confident ones) how would the detections from those two pruning strategies compare? Do the least confident queries still give good OBBs just not very confidently? Or are they only giving rubish?
- There are no pruning baselines evaluated leaving the reader to wonder about some of the algorithm choices and how close to optimal this approach performs.
    - How important is the incremental pruning vs pruning down to the desired number of queries in one go?
    - How well does an oracle-based pruner perform? Always prune the worst query when matched against GT bounding boxes. How does that pruned model perform relative to the proposed score based on?
- I appreciate that the timings were collected on a Jetson Nano device which shows the clear motivation to deploy the algorithms in practice. Checking against related methods, though the timing information was typically performed on a desktop-grade GPU. Showing those timings in addition would allow more direct comparability. I think Table 3 and 4 could have been combined to make room for the aforementioned additional investigations.
- Showing reprojections of 3D bounding boxes into frames hides any depth errors. So I would have appreciated Fig 4 to include some top down 3D visualizations instead of just frame reprojections (which frankly all look very similar).
- Algo 1: showing an algorithm is meant to help clarify the proposed method. I do not think it is achieving this aim as is. I think the algorithm depiction should be simplified more to where it actually captures the high level information. The details can certainly be found in the code once it is released.

**Questions:**

Fig 2 renders poorly for me and the text is hard to read. Please fix.

---

> ### Author Response · Authors · 2024-11-23
>
> Thank you for your constructive feedback! We will go through each of your points one by one and provide our responses accordingly.
>
> # Response to Weakness
>
> 1. That is correct. To help readers better understand the difference between our method and training from scratch with fewer queries, we added Figure 4 in the updated version of the paper. Figure 4 illustrates that queries obtained by training from scratch with fewer queries are more scattered and disordered, whereas queries pruned using GPQ not only retain the same spatial span but are also more organized.
>     1. Pruned queries are tend to generate lower classification scores, which means these queries will predict background class. They can be removed because their results can be covered by other more confident queries as illustrated in Fig.1(a) in the paper. To visually demonstrate the differences in queries before and after pruning, we have plotted the reference points used to generate the queries, as shown in Figure 4 of the updated paper.
>  This figure provides a clearer view of the impact of query pruning: redundant queries are removed, while the remaining queries visibly cluster into a plane. The plane formed by these points represents the solution space. After pruning, the remaining points still occupy the same space as before, which allows the original model's performance to be preserved.
>
>     2. We try to prune top most confident queries(**pruning highest**) and also prune the queries just in one go(**prune in one iteration**), the results above show that the model hasn't been destroyed, but still suffers a perference decrease.
>
> Model|mAP$\uparrow$|NDS$\uparrow$|mATE$\downarrow$|mASE$\downarrow$|mAOE$\downarrow$|mAVE$\downarrow$|mAAE$\downarrow$
> :---|---|---|---|---|---|---|---
> StreamPETR baseline|37.83%|0.4734|0.6961|0.2822|0.6846|0.2856|0.2084
> GPQ|**39.42%**|**0.4941**|**0.6766**|**0.2711**|0.5799|**0.2780**|0.2136
> **prune highest**|34.34%|0.4563|0.7429|0.2813|0.5912|0.3188|0.2195
> **prune in one iteration**|35.71%|0.4677|0.7121|0.2828|0.5970|0.2930|0.2233
> **prune using cost**|38.78%|0.4899|0.6808|0.2814|**0.5791**|0.2853|**0.2130**
>
> 2. Thank you very much for your suggestion! Exploring pruning strategies is indeed necessary. We conducted experiments following your suggestion:
>     1. The results of pruning down to the desired number of queries in one go are shown above, which evaluates that pruning queries gradually is better than pruning them all in one iteration.
>
>     2. We conduct the experiment that uses cost generated by assigner as the pruning metric. The results are shown above. By using cost instead of classification scores, the model could maintain its performance but is still worse than pruning using classification scores. This is because, during inference,  the only metric to justify  which query would be selected is classification socre. No assigner cost would be generated because there is no gt during inference. So the classification score is more suitable for acting as the pruning metric.
>
> 3. 3D object detection is a highly practical research area with the ultimate goal of deployment on various edge devices. Therefore, we chose Jetson Nano to test its performance, aiming to closely approximate its behavior in real-world production environments. However, our method also demonstrates strong performance on desktop GPUs. The anonmyous link https://github.com/Anonymity2025/files/blob/main/main_results.md provides the results of GPQ on a single RTX 3090. Results show that GQP can also  accelerate model inference on desktop GPUs. For example, in the case of PETR, pruning from 900 to 150 queries results in an mAP of 30.52\%,  which is 2.15 points higher than training from scratch with 150 queries (28.37\%), and its speed increases from 7.1 fps to 9.3, which is 1.31x faster
>
> 4. We have added top-down view to Fig.4. Please refer to Fig.5 in the revised PDF (pp.15) for detailed results.
>
> 5. Following your suggestion, we have simplified the algorithm (pp.6 in the revised PDF).
>
> # Response to Questions
>
> In the newly submitted PDF, we updated Figure 2 by improving its resolution, boldening the lines, and revising some text that might have caused misunderstandings.

---

> > ### Comment · Reviewer_EdfF · 2024-11-23
> > **Response**
> >
> > The authors response do address my concerns. It is good to see the comparison to other pruning strategies (which another reviewer also requested), as well as the visualization of the pruned queries vs from scratch queries in Fig 4 of updated paper.
> > Looking at the other reviewers and responses, I am happy to see that 2D DETR models can also profit from the same ideas.
> > I am leaning to upgrade my rating to weak accept, but would like to see the other reviewers responses as well.

---

> > > ### Author Response · Authors · 2024-11-26
> > >
> > > Thank you very much for your thoughtful feedback and for taking the time to review our responses. We are pleased to hear that our clarifications, including the comparison to other pruning strategies and the visualization of pruned vs. from-scratch queries in Figure 4, have addressed your concerns. We also appreciate your recognition of the broader applicability of our approach to 2D DETR models.
> > >
> > > We are grateful for your positive consideration of our work, and we hope that, after reviewing the feedback from the other reviewers, you will feel comfortable submitting an updated score in the system.
> > >
> > > Thank you again for your constructive comments, and we look forward to hearing your final thoughts.

---

### Official Review · Reviewer_Xnmm · 2024-11-03

**Soundness:** 2
**Presentation:** 3
**Contribution:** 2
**Rating:** 5
**Confidence:** 4

**Summary:**

The paper proposes query pruning in typical query-based object detectors to improve their performance. Experiments on the nuscenes dataset show the effectiveness of the approach.

**Strengths:**

+ The idea of using classification scores to prune queries is nice.
+ The paper is easy to understand.
+ The approach demonstrates promising results on the nuScenes validation set.

**Weaknesses:**

- An alternate way to reduce the processing time of 3D detectors is to use Token Merge [A]. How does query prune quantitatively compare against Token Merge [A]?
- Another way to reduce processing time is to use model quantization. How does query pruning quantitatively compare against Model Quantization?
- It would be beneficial to quantitatively include results from the nuScenes leaderboard, particularly comparing against a strong camera baseline like SparseBEV with 640x1600 resolution.
- The experiments are conducted with super small backbone (ResNet50). It would be insightful to quantitatively evaluate the approach on higher resolutions (512x1508 and 640x1600) to assess its performance.

References:
- [A] Token Merging: Your ViT but faster, Bolya et al, ICLR 2023.

**Questions:**

Please see the weakness

---

> ### Author Response · Authors · 2024-11-24
>
> We are grateful for your thoughtful comments and will provide answers to your queries in the order presented.
>
> 1. The comparison with ToMe is indeed very meaningful. We applied TokenMerge(ToMe) to StreamPETR to compare with GPQ. Results are listed in table below, which shows that ToMe cannot work on 3D object detection. ToMe not only failed to accelerate the model's inference but also caused the model to run slower.
>     - ToMe aims to prune ViT models. There are significant differences between image classification and 3D object detection methods, with the most notable being the number of tokens. Image classification methods typically require about 200 tokens, whereas in 3D object detection, due to the presence of surround views, the number of tokens can easily exceed 4000, with some methods generating more than 16,000 tokens.
>     - ToMe divides $N$ tokens into two groups with equal size $\frac N2$, then calculate a matrix of shape $\frac N2\times \frac N 2$ to merge $r$ most similar token pairs. For ViT methods, this matrix is less than 200x200 (about 200 tokens with patch size 16x16). However, for 3D detection methods, the matrix can easily larger than 2000x2000 (for example, 4224 tokens in StreamPETR and FocalPETR with 704x256 image size ) and can even reach 8000x8000 (16896 tokens in PETR with 1408x512 image size), which is 100 to 400 times larger than in ViT! The immense token size causes Token Merge to consume a significant amount of computational resources just to compute the similarity matrix when applied to 3D object detection. Consequently, instead of speeding up the computation, it actually makes the model run slower, as shown in the table.
>
> Model|r|Memory(MiB)|Speed(fps)|mAP|NDS|mATE|mASE|mAOE|mAVE|mAAE
> :--|--:|:---:|:---:|---:|---|---|---|---|---|---
> StreamPETR|-|2332|16.1|37.83%|0.4734|0.6961|0.2822|0.6846|0.2856|**0.2084**
> StreamPETR-ToMe|2112|2340|16.0|31.96%|0.4325|0.7546|0.2907|0.6893|0.3170|0.2210
> StreamPETR-ToMe|1056|2340|15.4|36.34%|0.4608|0.7187|0.2852|0.6963|0.2965|0.2121
> StreamPETR-ToMe|528|2338|14.9|37.69%|0.4721|0.6994|0.2822|0.6855|0.2877|0.2088
> StreamPETR-ToMe|264|2340|14.8|37.79%|0.4731|0.6982|0.2819|0.6849|0.2849|0.2084
> StreamPETR-GPQ|-|**2324**|**18.7**|**39.42%**|**0.4941**|**0.6766**|**0.2711**|**0.5799**|**0.2780**|0.2136
>
> 2. Our approach is not intended to replace model quantization methods. In fact, after applying our method, it is entirely possible to simultaneously use int8/int4 quantization to compress the model further. However, model quantization is highly sensitive to hardware, and modern machine learning models often rely on numerous third-party libraries (e.g., Flash Attention). This poses significant challenges for model quantization. When deploying the same model across different hardware platforms, it often requires hardware-specific adaptations, such as restructuring and removing certain components, followed by retraining. This process can lead to performance degradation and instability.
>
>     In our experiments, we attempted to quantize the model. However, we encountered difficulties running the quantized model on GPUs, which forced us to run the model on CPU. The comparison of performance after quantization is provided below.
>
> Model|Queries|mAP|NDS|mATE|mASE|mAOE|mAVE|mAAE
> :---:|:---|:---:|:---:|:---:|:---:|:---:|:---:|:---:
> PETR|900/-|31.74%|0.3668|0.8395|0.2797|0.6153|0.9522|0.2322
> PETR-int8|900/-|31.54%|0.3657|0.8483|0.2794|0.6084|0.9515|0.2326
> PETR-GPQ|900/300|32.85%|0.3884|0.8003|0.2791|0.5507|0.9108|0.2179
>
>   The FPS is not reported because the CPU speed is too slow to be meaningful.
>
> 3. It is essential to validate the effectiveness of GPQ on well-known methods from the nuScenes leaderboard. We can provide the results of GPQ on SparseBEV with an image resolution of 704x256, which shows that GPQ works well for SparseBEV. However, for a resolution of 1600x640, unfortunately, we are unable to run the experiments due to the limited time and computational power of our GPU (RTX 3090). In fact, even with a low resolution (704x256), running SparseBEV remains computationally expensive for us.
>
> Model|Backbone|ImageSize|Queries|FPS|mAP|NDS|mATE|mASE|mAOE|mAVE|mAAE
> :---:|:---:|:---:|:---|---|:---:|---|---|---|---|---|---
> SparseBEV|ResNet50|704x256|900/-|-|45.44%|0.5559|0.5980|0.2705|0.4141|0.2437|0.1866
> SparseBEV|ResNet50|704x256|900/300|-|43.28%|0.5428|0.5943|0.2738|0.4300|0.2488|0.1887
>
>   - If you want to examine the impact of higher resolutions on our method, you can refer to the PETR experiments conducted with VoVNet at a resolution of 1600x640 (see anonmyous link https://github.com/Anonymity2025/files/blob/main/petr_vov_1600x640.md). It took us over a week to run the experiments, and we would greatly appreciate it if you could consider these results.
>
> 4. We provide results of PETR using VovNet as backbone on higher resolution 1600x640 (see anonmyous link above), which proves that GPQ can work on large backbone and higher image resolutions.

---

> ### Author Response · Authors · 2024-11-26
>
> Dear Xnmm,
>
> We hope this message finds you well.  We are grateful for the time and effort you have spent reviewing our work, and we hope that our clarifications have addressed your concerns adequately.
>
> If you have any further questions or require more information on any aspect of our submission, please don't hesitate to let us know. We would be happy to provide any additional details or clarification.
>
> Once again, thank you for your valuable feedback. We look forward to hearing your thoughts.

---

> ### Comment · Reviewer_Xnmm · 2024-12-01
> **Response to Rebuttal**
>
> Thank you for the rebuttal. I read response from other reviewers' as well. However, I maintain my original score as the authors do not address the following concerns:
>
> - The paper does not compare performance with SparseBEV on the **nuScenes leaderboard** at 640x1600 resolution, even though SparseBEV releases their leaderboard model at 640x1600 resolution. This raises doubts about the effectiveness at higher resolutions, limiting its practical utility in cloud-deployments. It's **absolutely crucial** for 3D detection methods published in top-tier conferences like ICLR, NeurIPS, ICCV or CVPR to include nuScenes leaderboard results at **640x1600** resolution.
>
> - Thank you for providing PETR results. I appreciate these results, but, PETR is a very old detector, and is no longer a SoTA. The key application of the proposed method is to cut down the inference time of bigger resolution / SoTA leaderboard models such as SparseBEV at **640x1600** resolution.
>
> - AP and NDS for proposed method and SparseBEV even at 256x704 resolution take a significant hit of absolute 2 and 1 points respectively in newer results.

---

### Official Review · Reviewer_xxXP · 2024-11-04

**Soundness:** 2
**Presentation:** 1
**Contribution:** 2
**Rating:** 3
**Confidence:** 4

**Summary:**

This work proposes Gradually Pruned Queries (GPQ), which removes redundant queries in transformer-based 3D object detection models. Queries are sorted by classification scores, and the lowest-score query is iteratively removed. Experiments on the nuScenes dataset validate the effectiveness of GPQ.

**Strengths:**

- This work proposes a simple and reasonable pruning method for transformer-based 3D detection models.

**Weaknesses:**

- Numerous typos are present, such as "3D objec detection" (L351), "each query as as the fundamental" (L264), "it is the queries" (L268), "The reason of why our method" (L300). The authors are encouraged to correct all typographical errors and improve their English writing to enhance readability.
- Lack of comparison with prior work. This paper does not provide comparisons with previous pruning methods or implemented baseline pruning techniques. The authors are encouraged to include comparisons with other pruning baselines to demonstrate the advantages of their approach.
- No ablation studies. This paper lacks ablation studies on the design aspects of their method. The authors are encouraged to conduct ablation experiments to illustrate the impact of their design choices.

**Questions:**

- In Table 1, could the authors provide time and memory costs instead of accuracy? This would allow readers to more easily understand the relationship between the number of queries and model efficiency.
- In Figure 3, should the last column be labeled "Iteration 𝑛+1"? Or is Iteration 𝑛 run twice in this work?
- In Table 2, could the authors include model cost along with accuracy? This addition would help readers better understand the pruning effect of the proposed method.

---

> ### Author Response · Authors · 2024-11-23
>
> Thank you for your valuable insights. We will respond to your questions sequentially
>
> # Response to Weakness
>
> 1. We are sorry for typos, and we thoroughly reviewed the manuscript and corrected all identified typos and inconsistencies.
>
> 2. Our response to this comment is from two perspectives:
>     - First, comparison with previous work is challenging because, to our knowledge, no previous research has focused on query pruning. Traditionally, pruning in the visual domain has primarily been applied to models for image classification tasks, such as ViT, with very little work targeting 2D object detection and almost none targeting 3D object detection methods. There are significant differences between image classification, 2D object detection  and 3D object detection methods, the most notable being the number of tokens. Here, "tokens" refer to the features extracted from the input image by the image backbone. Image classification methods typically require only a few hundred tokens, 2D object detection methods use around 2000 tokens, while 3D object detection can easily exceed 4000 tokens due to the presence of surround views (StreamPETR, FocalPETR with image size 704x256). Some methods even generate more than 16,000 tokens (PETR, SparseBEV with image size 1600x640). We argue that the significant difference in the number of tokens makes it difficult for ViT pruning methods to perform effectively when applied to 3D object detection. For example, we conducted experiments using Token Merge (ToMe) [1], and the results can be found in an anonymous file  https://github.com/Anonymity2025/files/blob/main/tome.md. In the table, "StreamPETR-GPQ" is our method with 900 initial queries and 300 final queries; column r represents the number of tokens to "merge" as defined in the ToMe method. As indicated in ToMe, in order to merge tokens, we need to divide the tokens into two groups and then compute a similarity matrix of the form $\frac{N}{2}\times\frac{N}{2}$, where $N$ is the number of tokens. The calculation of the similarity matrix introduces additional computational overhead, which not only fails to speed up inference, but also slows down the model. This makes ToMe ineffective for 3D object detection methods.
>
>     - Second, due to differences in model architectures, we believe that many existing ViT pruning methods are not directly applicable to 3D object detection. In ViT, the tokens generated by the backbone serve as queries, keys and values, resulting in an attention map with a square matrix shape. This specific shape makes methods based on it unsuitable for the domain of 3D object detection [2]: in 3D object detection methods, the attention map is generated by predefined queries and keys whose numbers are not equal. As a result, the attention map is not a square matrix.
>
>     [1] Token Merging: Your VIT but Faster. ICLR 2023.
>
>     [2] Zero-TPrune: Zero-Shot Token Pruning through Leveraging of the Attention Graph in Pre-Trained Transformers. CVPR 2024.
>
> 3. We agree that ablation experiments are important. To validate the effectiveness of our method, we complement the following ablation experiments:
>     1. **Reasonability of pruning criteria**: Our pruning strategy involves removing the queries with the lowest classification scores. To demonstrate the validity of this criterion, we designed two experiments: one where we prune the queries with the highest classification scores(**prune highest**), and another where we guide the pruning using the cost generated by the assigner during the binary matching process between predicted and ground truth values(**prune using cost**) .
>
>     2. **Reasonability of the gradual pruning strategy**: We designed an experiment that pruned 600 queries in a single iteration to demonstrate that our  gradual pruning strategy is reasonable and effective (**prune in one iteration**).
>
> Model|mAP$\uparrow$|NDS$\uparrow$|mATE$\downarrow$|mASE$\downarrow$|mAOE$\downarrow$|mAVE$\downarrow$|mAAE$\downarrow$
> :---|---|---|---|---|---|---|---
> StreamPET Rbaseline|37.83%|0.4734|0.6961|0.2822|0.6846|0.2856|0.2084
> GPQ|**39.42%**|**0.4941**|**0.6766**|**0.2711**|0.5799|**0.2780**|0.2136
> **prune highest**|34.34%|0.4563|0.7429|0.2813|0.5912|0.3188|0.2195
> **prune in one iteration**|35.71%|0.4677|0.7121|0.2828|0.5970|0.2930|0.2233
> **prune using cost**|38.78%|0.4899|0.6808|0.2814|**0.5791**|0.2853|**0.2130**
>
>   The results show that pruning using the highest classification scores, pruning based on assigner cost, or pruning a large number of queries in a single step all degrade the model's performance after pruning. This demonstrates that the strategy designed in GPQ is currently the optimal query pruning strategy.
> # Response to Questions
> 1. We have updated the Table 1, adding memory costs. Details can be found in updated PDF (pp.2).
> 2. Thanks for pointing this mistake. It should be iteration $n+1$, and we have fixed it.
> 3. We have updated the Table 2, where following your suggestions, we newly added models' costs.

---

> ### Author Response · Authors · 2024-11-26
>
> Dear xxXP:
>
> We hope you're doing well. We truly appreciate the time you've dedicated to reviewing our work, and we hope the clarifications we offered have addressed your concerns satisfactorily.
>
> If there are any further questions or aspects that you would like us to elaborate on, we would be glad to offer more information.
>
> Thank you once again for your insightful feedback, and we look forward to receiving your thoughts.

---

### Official Review · Reviewer_nz4D · 2024-11-04

**Soundness:** 2
**Presentation:** 3
**Contribution:** 2
**Rating:** 5
**Confidence:** 3

**Summary:**

In query-based models, the number of queries significantly exceeds the actual number of objects for both 2d object detection and 3d object detection. The performance of models is positively correlated with the number of queries, and the performance consistently declines as the number of queries is reduced. In this paper, the authors propose that queries do not contribute equally and even there are queries are never selected as final results. The redundant queries result in unnecessary computational and memory costs. Therefore, the authors propose a query pruning method to gradually prune redundant queries in transformer.

**Strengths:**

1. This paper is the first study to explore query pruning in query-based detectors.
2. The experiments demonstrate the effectiveness of proposed method, which reduces redundant queries while maintaining performance.

**Weaknesses:**

1. The experiment is incomplete and unpersuasive.

**Questions:**

1. The font in Figure 1 is not consistent with other figures. The label of Y-axis is not clear.
2. In query-based models, there are content query and spatial query. Does the proposed method perform experiments on the detectors that use both content query and spatial query? If not, the author should add additional experiments to prove the effectiveness of the proposed method on this kind of detectors.
3. In StreamPETR and FocalPETR, two variants of the model are trained for 24 epochs and 60 epochs, respectively. In StreamPETR, the models in the ablation study are trained for 24 epochs, while trained for 60 epochs when compared with others. The authors should add additional comparison experiments which trained for 60 epochs.
4. For model pruning methods, except for precision, FLOPs and time, the number of parameters and memory cost are also important evaluation metrics. The authors should provide the comparison experiments between detectors based on the number of parameters and memory cost.

---

> ### Author Response · Authors · 2024-11-24
>
> We sincerely appreciate your feedback and will address each of your comments in the order they were raised.
>
> # Response to Weakness
>
> Due to time and computational resource constraints, our initial experiments were not comprehensive. We have since supplemented them with additional experiments:
>
> 1. We evaluate more models, including DETR3D. You can check the results via anonmyous links https://github.com/Anonymity2025/files/blob/main/main_results.md and https://github.com/Anonymity2025/files/blob/main/sparsebev.md. Results have show that GPQ works well on DETR3D.
> 2. We test the inference speed on a desktop GPU (RTX3090), showing that our method can accelerate model inference by up to 1.31x.
> 3. We change models' backbone, use higher resolution (1600x640). Results are listed in anonymous file https://github.com/Anonymity2025/files/blob/main/main_results.md, which shows that GPQ still works when using larger backbone and higher resolution.
> 4. We also apply our method to ConditionalDETR. Results are listed in anonmyous file https://github.com/Anonymity2025/files/blob/main/conditional_detr.md, which shows that GPQ also works for 2D object detection.
> 5. We compare GPQ with an existing pruning method Token Merge(ToMe)[1], which shows that GPQ is better than ToMe. Results are listed in table below:
>
> Model|$r$|Memory(MiB)|Speed(fps)|mAP|NDS|mATE|mASE|mAOE|mAVE|mAAE
> :---|---:|:---:|:---:|---:|---|---|---|---|---|---
> StreamPETR|-|2332|16.1|37.83%|0.4734|0.6961|0.2822|0.6846|0.2856|**0.2084**
> StreamPETR-ToMe|2112|2340|16.0|31.96%|0.4325|0.7546|0.2907|0.6893|0.3170|0.2210
> StreamPETR-ToMe|1056|2340|15.4|36.34%|0.4608|0.7187|0.2852|0.6963|0.2965|0.2121
> StreamPETR-ToMe|528|2338|14.9|37.69%|0.4721|0.6994|0.2822|0.6855|0.2877|0.2088
> StreamPETR-ToMe|264|2340|14.8|37.79%|0.4731|0.6982|0.2819|0.6849|0.2849|0.2084
> StreamPETR-GPQ|-|**2324**|**18.7**|**39.42%**|**0.4941**|**0.6766**|**0.2711**|**0.5799**|**0.2780**|0.2136
>
>
> In the table, column $r$  represents the number of tokens to be "merged" as defined in the ToMe method.
>
> We have incorporated these revisions into the updated paper. Newly added and revised content is highlighted in red. We will be very grateful if you could refer to the updated PDF of the paper for full details.
>
> [1] Token Merging: Your VIT but Faster. ICLR 2023.
>
>
> # Response to Questions
>
> 1. We have fixed Fig.1 in updated PDF.
> 2. Yes, we use both content query and spatial query. In fact, in all of transformer-based 3D detection methods, the content query and spatial query are both used. Content query just is the "pre-defined query"  mentioned in our paper, and the "spatial query" is usually called "positional embedding", which will be added to the "content query". The positional embedding is calculated using pre-defined query, so we just need to prune pre-defined queries.
> 3. Due to resource limitations (our experimental resources are two RTX 3090 GPUs), it is challenging for us to support longer training durations in a short time. However, we still conducted experiments with more epochs (36 epochs, **StreamPETR-36e** in the table below), and the results are shown in the table below. Here **StreamPETR-GPQ** loads the checkpoint generated by StreamPETR-36e.
>
> Model|Queries|mAP$\uparrow$|NDS$\uparrow$|mATE$\downarrow$|mASE$\downarrow$|mAOE$\downarrow$|mAVE$\downarrow$|mAAE$\downarrow$
> :---:|:---|:---:|:---:|:---:|:---:|:---:|:---:|:---:
> StreamPETR-24e|900/-|37.83%|0.4734|0.6961|0.2822|0.6846|0.2856|0.2084
> StreamPETR-36e|900/-|**41.34%**|0.5110|0.6538|**0.2744**|0.5580|**0.2710**|**0.1999**
> StreamPETR-GPQ|900/300|41.00%|**0.5144**|**0.6410**|0.2750|**0.5050**|0.2833|0.2022
>
>   The experimental results shown in the table further validate that our method remains effective even when applied to models trained for more iterations. After pruning redundant queries, the model is still able to maintain its performance.
>
> 4. We fully agree with your point that the number of parameters and memory usage are also important metrics for pruning methods. Pruning queries has minimal impact on the model's parameters and memory usage, as the number of parameters associated with queries in the transformer decoder is inherently small. Details are provided in the table below. The primary benefit of pruning queries lies in improving the model's inference speed (see anonmyous file https://github.com/Anonymity2025/files/blob/main/main_results.md).
>
> Model|Backbone|ImageSize|Queries|Parameters|Memory(MiB)
> :---|:---:|:---:|:---|---:|---:
> PETR|R50|1408x512|900/-|36670547|3070
> PETR|R50|1408x512|900/300|36668747|2968
> PETR|R50|1408x512|900/150|36668297|2732
> PETR|VOVnet|1600x640|900/-|81664736|5678
> PETR|VOVnet|1600x640|900/300|81662936|5352

---

> ### Author Response · Authors · 2024-11-26
>
> Dear nz4D,
>
> We hope this message finds you well.  We appreciate the time and effort you've dedicated to reviewing our work, and we hope that our clarifications address your concerns effectively.
>
> If you have any further questions or if there are any additional points you would like us to elaborate on, we would be more than happy to provide additional information.
>
> Thank you again for your valuable feedback, and we look forward to your thoughts.

---

### Author Response · Authors · 2024-11-26

# To all reviewers

First, please allow us to express our gratitude to all reviewers. In this paper, we propose a pruning method for Query-based 3D object detection approaches, aiming to prune redundant queries in the model.

We discussed the redundancy of queries in existing 3D object detection methods in the paper and proposed a gradual pruning approach that eliminates redundant queries while maintaining the performance of the original model.

- **Reviewer nz4D** suggested that we should add results of applying GPQ to base models trained for more epochs.
- **Reviewer xxXP** recommended including ablation studies and comparisons with other methods.
- **Reviewer Xnmm** suggested comparing GPQ with the ToME method and model quantization approaches, and adding experimental results on models with larger backbones and image sizes.
- **Reviewer EdfF** raised questions about query pruning strategies and metrics. Based on their guidance, we added a comparison of model runtime performance before and after pruning on desktop GPUs.
- Following **Reviewer KmgM**'s suggestion, we extended GPQ to the 2D object detection domain and explored the integration of GPQ with training, allowing for pruning during training.


# List of changes

According to the comments of all reviewwers, we have made the following changes to our paper:

1. In **Table 1**, we have added the memory usage and FPS changes with varying query numbers (Page 2).
2. In **Section 2.2**, we have explained why existing pruning methods for Transformers cannot or are difficult to transfer to 3D object detection models (Page 4).
3. In **Section 3.4**, we have inserted a new **Figure 4** to more clearly demonstrate the distribution differences between queries pruned using GPQ and those obtained by training with fewer queries from scratch (Page 7).
4. In **Table 2**, we have added more models and included backbone, image size, and model runtime (FPS) on RTX3090 (Page 8).
5. In **Section 4.3.2**, we merged the original **Table 3** and **Table 4** (Page 9).
6. We have added **Section 4.3.3** (Page 9), where we compare GPQ with previous methods.
7. We have added **Section 4.4** (Page 10) to include ablation experiments.
8. We moved the original **Figure 4** to the appendix (now **Figure 5**, Page 15) and added visualizations from a top-down view.
9. In **Appendix A.2**, we have added the effects of GPQ in 2D object detection (Page 16).
10. In **Appendix A.3**, we have added the resules of integrating GPQ with training from the beginning (Page 16).
11. In **Appendix A.4**, we have added the results of GPQ on fully converged models (Page 17).
11. Additionally, we made several minor adjustments, such as changes to the font in images, fixing typos, and increasing image resolutions.

---

### Note · Authors · 2024-12-14

I have read and agree with the venue's withdrawal policy on behalf of myself and my co-authors.